# A Comprehensive 2D-LC/MS/MS Profile of the Normal Human Urinary Metabolome

**DOI:** 10.3390/diagnostics12092184

**Published:** 2022-09-09

**Authors:** Jiyu Xu, Shuxin Zheng, Mimi Li, Xiaoyan Liu, Haidan Sun, Zhengguang Guo, Jing Wei, Lulu Jia, Wei Sun

**Affiliations:** 1Core Facility of Instrument, Chinese Academy of Medical Sciences, School of Basic Medicine, Peking Union Medical College, Beijing 100005, China; 2Human Biology, Art and Science, University of Toronto, 27 King’s College Circle, Toronto, ON M5S 1A1, Canada; 3Clinical Research Center, National Center for Children’s Health, Beijing Children’s Hospital, Capital Medical University, Beijing 100054, China

**Keywords:** urinary metabolome, urinary metabolites, metabolomics database

## Abstract

Profiling bodily fluids is crucial for monitoring and discovering metabolic markers of disease. In this study, a comprehensive analysis approach based on 1D-LC-MS/MS and 2D-LC-MS/MS was applied to profile normal human urine metabolites from 348 children and 315 adults. A total of 2357 metabolites were identified, including 1831 endogenous metabolites and 526 exogenous ones. In total, 1005 metabolites were identified in urine for the first time. The urinary metabolites were mainly involved in amino acid metabolism, small molecule biochemistry, lipid metabolism and cellular compromise. The comparison of adult’s and children’s urine metabolomes showed adults urine had more metabolites involved in immune response than children’s, but the function of binding of melatonin, which belongs to the endocrine system, showed a higher expression in children. The urine metabolites detected by the 1D-LC-MS/MS method were mainly related to amino acid metabolism and lipid metabolism, and the 2D-LC-MS/MS method not only explored metabolites from 1D-LC-MS/MS but also metabolites related to cell signaling, cell function and maintenance, etc. Our analysis comprehensively profiled and functionally annotated the metabolome of normal human urine, which would benefit the application of urinary metabolome to clinical research.

## 1. Introduction

A dynamic response by biological systems to various biological conditions, including disease and exposure to the environment, can be revealed by low-molecular-weight metabolic analytes and their intermediates [1,2]. Metabolic analytes are the products of gene expression and protein activity [3,4]. Additionally, they are composed of various chemical compounds, including sugars, amino acids, nucleic acids, lipids, simple fatty acids, etc. [5,6]. Urine is one of the most commonly used biological samples in metabolome research, which has many advantages for clinical research, such as easier collection, larger quantity, lower protein amounts and high amounts of metabolites [7]. Urine-based metabolites measurement has been a routine practice in the search for the biomarkers of human diseases, such as obesity, colorectal cancer and cardiovascular disease [8,9,10].

A comprehensive profile of the normal urinary metabolome is essential in the biomarker discovery process. The technological advancement of mass spectrometry (MS) made it possible to identify a greater number of urinary metabolites. In 2011, Kevin et al. detected 3564 13C-/12C-dansylated ion pairs or metabolites in a human urine sample using the 2D-LC–MS strategy [11]. In 2013, Souhaila et al. reported the human urine metabolome (UMDB) containing 2651 human urine metabolites, in which 445 metabolites were identified by multi-platform metabolomic analysis and 2206 urinary compounds were determined from a literature review [7]. During the last few years, there has been much effort put into identifying more urinary metabolites [12]. Currently, the Human Metabolome Database (HMDB) contains 7238 urinary metabolites identified by NMR, GC-MS, DFI/LC-MS/MS, ICP-MS and HP/LC-MS/MS [13]. Technological advancements have allowed for the establishment of a urine metabolome database. Additionally, a deeper and more comprehensive database may be helpful in urine metabolomic research. 

In this study, to achieve optimal coverage of urine metabolomes, one- and two-dimensional separation strategies (Figure 1) were employed to profile urine derived from 348 healthy children and 315 healthy adults. A readily available source for the human urinary metabolome, the “Human Urinary Metabolome Database,” could be used for in-depth analysis. We compare the urine metabolomic differences not only between children and adults but also between the 1D LC-MS/MS and 2D LC-MS/MS methods. Function and pathway annotation of urine metabolites was further conducted to explore metabolite functions. A urine metabolome dataset can provide insights into urine metabolite physiological function and be used for biomarker discovery.

## 2. Materials and Methods

### 2.1. Urine Collection 

Informed consent was obtained from all human subjects before participation in our study. This study was approved by the Ethics Committee for Human and Animal Research in Peking Union Medical College (no. 047-2019) regarding the consent process and the research protocol. The study methodologies were in accordance with the Helsinki Declaration (as revised in 2013). For minors (age < 18), consent was obtained from their guardians. All healthy participants were checked and examined by trained doctors according to standard operating procedures. The inclusion criteria were as follows: (1) clinical laboratory tests indicating no acute or chronic disease, (2) no recent use (<2 weeks) of any drugs, (3) a qualified physical examination finding no dysfunction of vital organs and (4) normal renal function. Strict exclusion criteria were applied: (1) with any urinary system disease, (2) with any types of genetic diseases, (3) on a restricted diet and (4) pregnant or breastfeeding women. Early morning urine (midstream) was collected from a cohort of 315 clinically healthy adults and 348 clinically healthy children on an empty stomach between 7:00 and 9:00. Their detailed demographics are shown in Appendix A. 

### 2.2. Urine Sample Preparation 

Acetonitrile (Thermo Fisher Scientific, Waltham, MA, USA) was added to each urine sample (1:2, *v*/*v*), then the mixture was vortexed for 30 s and centrifuged at 14,000× *g* for 10 min. The supernatant was vacuum-dried and then redissolved in 2% acetonitrile. Next, 10 KDa molecular weight cut-off ultra-centrifugation filters (Millipore Amicon Ultra, Billerica, MA, USA) were used to separate larger molecules from urinary metabolites before transferring them to the autosamplers. The final samples of children and adults were a pooled urine sample prepared by mixing aliquots of 315 adult samples and 348 child samples. Due to the large sample size, a bulk QC sample can be prepared from a representative subset of subjects [14]. The quality control (QC) sample was a pooled urine sample prepared by mixing aliquots of 50 representative samples across different groups to be analyzed and was therefore globally representative of the whole sample set. The QC samples were injected every 10 samples during the analytical run to assess the stability and repeatability of the method. The 348 children metabolites samples and 315 adult metabolites samples were separately pooled with equal amounts of metabolites into one sample for one- and two-dimensional analyses.

### 2.3. HPLC Separation

In this study, the metabolite mixtures were dried under vacuum, redissolved in 2% acetonitrile and fractionated with an HPLC column from Waters (4.6 mm × 250 mm, Xbridge C18, 3 μm). Each metabolite mixture was loaded onto the column in H_2_O (pH = 10). The elution gradient was set from 5% to 30% buffer B2 (90% ACN, pH = 10; with a flow rate = 1 mL/min) for 30 min. Eluted metabolites were collected as fractions per minute. The 30 fractions were vacuum-dried and then resuspended in 2% acetonitrile. All 30 fractions were pooled into 15 samples by combining fractions 1 and 16, 2 and 17, 3 and 18 and so on. A total of 15 fractions from one sample were analyzed by LC−MS/MS.

### 2.4. LC-MS/MS Analysis 

The detailed online LC-MS/MS were as follows. Ultra-performance LC-MS analyses of urine samples were performed on a Waters ACQUITY H-class LC system coupled with an LTQ-Orbitrap Velos mass spectrometer (Thermo Fisher Scientific, Waltham, MA, USA). Urinary metabolites were separated with a 29 min elution gradient on a Waters HSS C18 column. The flow rate was set to 0.3 mL/min. The A mobile phase was 0.1% formic acid in H_2_O, and acetonitrile was the B mobile phase. The elution gradients, 0–2 min with 2% buffer B, 2–5 min with 2–55% buffer B, 5–15 min with 55–100% buffer B, 15–20 min with 100% buffer B, 20–20.1 min with 100–2% buffer B and 20.1–29 min with 2% buffer B, were applied. The temperature of the column was set to 50 °C.

The MS acquisition mode was set to full scan from 100 to 1000 *m*/*z* at a resolution of 60,000. The automatic gain control (AGC) target was set to 1E6, and the maximum injection time was set to 100 ms. For UPLC targeted-MS/MS analyses, the settings were set as follows: the resolution was 15,000, AGC target was 5 × 10^5^, maximum injection time was 50 ms and isolation window was 3 *m*/*z*. There were three collision energies optimized for each target: 20, 40 and 60, with higher-energy collisional dissociation (HCD). A total of 1008 LC-MS/MS runs were conducted, including 547 runs for the urine samples of adults and 461 runs for the samples of children.

### 2.5. Data Processing

We used a previously described strategy to process the data [13]. Progenesis QI (Waters, Milford, MA, USA) software was used to process the raw data files. The detailed workflow for QI data processing and metabolite identification was described in a previous study [15]. Multivariate statistical analysis and compound confirmation were conducted after the identification results were exported. The score and fragmentation score provided by Progenesis QI were used to confirm the compounds. Each identification was scored from 0 to 60 based on its reliability. A fragment score threshold of 20.0 was set based on reference standard scores. The identified metabolites were screened by package R to distinguish endogenous and exogenous metabolites. 

Metabolic pathways were analyzed using the “Mummichog” algorithm based on the MetaboAnalyst 3.0 platform. The Ingenuity Pathway Analysis (Ingenuity Systems, Mountain View, CA, USA) tool was used for further functional classification of the identified urinary metabolites.

## 3. Results

### 3.1. Comprehensive Identification of Urinary Metabolome

In this study, one-dimensional and two-dimensional separation strategies were used to achieve maximal coverage of urinary metabolomes. Pooled urine samples derived from children and adults were used to establish a large urinary metabolites database. In children/adult groups, 1496/1691 metabolites were identified, including 1180/1334 from endogenous and 316/357 from exogenous, respectively. The resulting human urine database contains 2357 metabolites, including 1831 from endogenous (Table 1 and Appendix A) and 526 from exogenous (Table 2 and Appendix A).

The compounds identified in this study covered a range of chemical classes. Defined in HMDB as a chemical “Sub Class,” the top three metabolite sub classes were: amino acids, peptides, and analogues (32%), carbohydrates and carbohydrate conjugates (16%) and fatty acids and conjugates (9%) (Figure 2A), which is similar to a previous study [11]. Functional classifications and pathway analyses were carried out for the metabolites using the Ingenuity Pathway Analysis (IPA) tool (http://www.ingenuity.com/) and MetaboAnalyst 3.0. (Appendix A). Pathway analysis annotated a wide range of urine metabolites involved in amino acid metabolism, such as phenylalanine metabolism, tyrosine metabolism, arginine biosynthesis, etc. (Figure 2B). Function annotation showed that the detected metabolites were mainly involved in small molecule biochemistry, lipid metabolism and cellular compromise (Figure 2C).

Amino acids metabolites are the most important components of urine, which display remarkable metabolic and regulatory versatility. They are essential precursors for the synthesis of a variety of molecules of utmost importance and also regulate key metabolic pathways and processes that are crucial to the health, growth, development, reproduction and homeostasis of organisms [16]. Amino acids have an effect on the pathology, prognosis and treatment of many diseases, which has made them the focus of current clinical research. A previous study developed a capillary electrophoresis-tandem mass spectrometry (CE-MS/MS) method to quantify the amino acids in urine. Additionally, the profiling of amino acids established by the CE-MS/MS method was applied in clinical research on inflammatory bowel disease [17]. Another study found altered concentrations of some amino acids in prostate cancer patient urine. The results indicated urine amino acid metabolites were potential sources of prostate cancer biomarkers [18]. 

When comparing the identified metabolites in this study with HMDB, 1005 metabolites were not reported by HMDB (Figure 3A). In this study, the top sub class of annotated metabolites were amino acids, peptides and analogues, which were different from HMDB (fatty acids and conjugates) (Figure 3C). According to the IPA pathway analysis, our results showed higher expression levels of nucleosides and nucleotide biosynthesis and degradation, hormones degradation and amino acids degradation, which indicated that our results were complementary to HMDB (Figure 3D).

Urinary metabolites were considered to speak to the metabolite composition of the yield of the kidneys and were basically composed of metabolites determined from urinary tract system secretion and plasma filtration. Meanwhile, a few of the metabolites from the intestine are retained in the circulation and eventually chemically modified (that is, co-metabolized) by the host, then finally excreted with the urine [19,20]. A comparative analysis of the plasma, urine, feces metabolome would give a more concrete connection of plasma, urine and feces metabolome. The metabolites identified in this study were combined with HMDB urine metabolites to obtain a database containing 8243 urine metabolites. Comparing the combined urine database with the HMDB blood and fecal database, we found that 1059 metabolites co-existed with urine, blood and feces (Figure 3B). The co-existed metabolites were involved in a wide variety of metabolic functions, while metabolites unique to urine mostly enriched in Bile Acid and Cholesterol Biosynthesis, Amino Acid metabolism; metabolites unique to blood were involved in energy metabolism, oxidation—reduction and other response; and metabolites unique to feces mainly participant in glucose metabolism. 

### 3.2. Comparison Pathway and Functions of Metabolites Derived from Children and Adults

A total of 1180 metabolites were detected in the children’s urinary metabolome, and 1334 were detected in the adult’s urinary metabolome. The overlap between the children and adults is shown in Figure 4A. The metabolites detected in adults and children had similar chemical classifications (Figure 4B). IPA annotation showed that the urinary metabolites were mainly related to cellular processes, metabolism and signaling, information storage and processing and other processes (Figure 4C, Appendix A). According to IPA analysis, the expression of inflammatory response in adults was higher than that in children, while the function expression levels of amino acid metabolism, carbohydrate metabolism, energy metabolism and lipid metabolism were similar between children and adults. Additionally, IPA analysis showed that the binding of melatonin function was downregulated in adults.

### 3.3. Comparison of Two Separation Strategies

Two separation strategies were employed to profile urinary metabolites. For 1D separation, after sample preparation, samples were directly separated by online liquid chromatography with gradient elution, and then the coeluting analytes entered the mass spectrometer for further analysis. For 2D separation, offline liquid chromatography separation was used after sample preparation, and the coeluting analytes were collected per minute. A total of 30 fractions were collected; after that, the fractionated samples were pooled into 15 samples by combining fractions 1 and 16, 2 and 17, 3 and 18 and so on. Finally, 15 runs were carried out by online LC-MS/MS, which was the same as 1D separation. Representative chromatograms are shown in Appendix A. For 1D separation, the sample ran three times to increase the metabolome coverage. For 2D separation, 15 pooled fractions were analyzed, respectively, and the identification results were merged.

Among the 1831 endogenous metabolites, 540 metabolites were identified in the 1D analysis, and 1781 metabolites were identified in the 2D analysis. The overlap between the 1D and 2D analyses is shown in Figure 5A. Among the identified metabolites, 97.3% of the metabolites found in one-dimensional separation were also detected in two-dimensional separation, and 2.7% were uniquely found in one-dimensional separation. 

The number of metabolites of the top three sub classes identified in the 1D method was similar to that in the 2D method (Figure 5B). Function annotations of urinary metabolites based on the degree of analysis depth may be helpful in providing insight into the analysis approach difference in determining suitable separation methods for specific purposes. IPA functional annotation showed that the detected metabolites were involved in various processes, such as substance metabolism, cellular behavior, organismal functions and so on (Figure 5C, Appendix A). Regarding the comparison analysis, both 1D and 2D methods can detect metabolites related to amino acid metabolism and lipid metabolism. However, the 2D method detected more metabolites involved in other processes, such as cell signaling, cell function and maintenance, cellular assembly and organization. These findings can help us select proper separation methods for different research purposes. 

### 3.4. Identification of Metabolites Derived from Exogenous

Physiological status and pathological changes in an individual can be captured by a metabolic state that reflects the influence of both genetic variants and environmental factors such as diet, lifestyle and gut microbiome [20]. Therefore, the detection of exogenous metabolites is as important as the detection of endogenous metabolites. In this study, a total of 526 metabolites derived from exogenous were detected. The overlap among the metabolites identified in the children and adult analyses is displayed in Figure 6A. These compounds are mainly derived from three sources: food (317), environment (7) and synthesis (1) (Figure 6B). 

### 3.5. Websites of Human Normal Urine Metabolome

To assist in the navigation of the normal human urine metabolome, all of the mapped results can be easily searched and freely downloaded via http://www.urimarker.com/mdStudyAbsBrowse?&pageStart=0 (accessed on 21 August 2022) [21]. The Chinese human normal urinary metabolome database contains 2357 nonredundant metabolites, including 1831 endogenous metabolites and 526 exogenous ones. Detailed metabolites information, including the HMDB ID, the location, the chemical class and the source, is provided, which provides the basic reference dataset for future metabolome research.

## 4. Discussion

In this study, two separation strategies, one-dimensional and two-dimensional, were employed to profile normal human urine metabolites from 348 children and 315 adults. A human urinary metabolome database was established, including 1831 endogenous metabolites and 526 exogenous ones. In total, 1005 metabolites were identified in urine for the first time. Adults’ urine contained more metabolites involved in immune response than children’s urine, but the function of binding of melatonin showed a higher expression in children. As for the comparison of two separation methods, in addition to metabolites detected in both 1D and 2D separation, 2D separation detected more unique urinary metabolites.

According to a previous report, metabolite separation techniques before MS analysis can both reduce mass spectral complexity and provide additional information on metabolite physicochemical properties, which may help in metabolite identification [14]. The multidimensional liquid-based method could enhance the resolving power of chromatography and consequently, enhance the number of measurable metabolites, widen the overall dynamic ranges and increase the metabolome coverage [22,23]. Notably, an advanced workflow, HPLC × LC-MS, was employed in this study. The number of coeluting analytes entering the mass spectrometer ion source at any one time can be reduced, and ion suppression is decreased due to the up-front HPLC separation. Therefore, 2D separation acquired better mass spectral data quality, less background noise and detected more metabolites than 1D separation.

The urinary metabolome database established in this study includes 1005 newly identified metabolites. Specifically, it contains 132 metabolites from adults determined by 1D separation, 636 metabolites from adults determined by 2D separation, 177 metabolites from children determined by 1D separation and 534 metabolites from children determined by 2D separation (Appendix A). Most of the new metabolites (926/1005) were identified by 2D separation. A previous study employed multi-platforms, including NMR spectroscopy, gas chromatography-mass spectrometry (GC-MS), direct flow injection mass spectrometry (DFI/LC-MS/MS), inductively coupled plasma mass spectrometry (ICP-MS) and high-performance liquid chromatography (HPLC) to analyze 22 adult urine samples and established a urine metabolome database [7]. The previous study used multiple analysis approaches to provide a urine metabolome database. However, the previous analysis only used the 1D LC-MS approach; therefore, our 2D LC-MS could identify more metabolites. Moreover, the sample size was small, and a child sample was not included. In this study, we used a relative large-scale urine sample size, including 315 adults and 348 children, which would provide a more comprehensive urine database. 

Due to the large-scale sample size, including adults and children, we compared adult’s metabolomes with children’s metabolomes. Our results showed that urinary metabolites function had differences between the adult and child groups. The binding of melatonin function was found to be downregulated in adults by IPA analysis. Melatonin is secreted during darkness and plays a key role in different physiological responses, including the regulation of sleep homeostasis, circadian rhythms, vasomotor responses and retinal neuromodulation [24]. Melatonin decreases with age and has been reported in clinical settings and mice models [25]. Therefore, the urine metabolome could reflect the change in the endocrine system with increased age. 

Furthermore, the expression of inflammatory response in adults was higher than that in children, according to the IPA analysis. Inflammation is a hallmark of each major age-related disease and phenotype and has appeared to be a defining pathological characteristic of aging tissues across multiple species [26]. In this study, the expression level of many inflammatory-related functions in adults was higher than that in children, indicating that the functional expression of inflammatory-related metabolites increased with age. These changes may be due to adults having a higher content of metabolites involved in immune function, such as carnosine, creatine, cytidine, etc. It was reported that adults had higher creatine content in muscle than children [27]. Dietary supplementation with creatine could protect humans from infectious diseases by increasing the generation of nitric oxide [28]. Additionally, a previous study showed that muscle carnosine content increased from children to young adults and then decreased from adults to the elderly [29]. Overall, oral administration of carnosine reduced virus dissemination in humans [28].

However, there are some limitations to this study. First, all the participants were from a single center in northern China, and they had a similar lifestyle; therefore, the results may be population specific, and a large-scale analysis from multi-centers might achieve more comprehensive results. Second, potential factors such as smoking and other lifestyle and dietary habits might influence the metabolome. These factors should be considered in further study. Third, only nonpolar metabolites were analyzed in this study, and nonpolar metabolite analysis might enhance metabolome coverage. Last, all the analytes in this study were identified by matching the fragment to an online database. A standard spectral library might increase the confidence in metabolites identification.

## 5. Conclusions

In this study, we established a comprehensive profile of human normal urine metabolome utilizing LC/MS/MS. A total of 2357 metabolites were identified. A total of 1005 metabolites were reported as normal human urinary constituents for the first time. The comprehensive metabolome annotation of children and adults showed the similarity and differences in their biological pathways and functions. From a clinical point of view, this database provides researchers of the metabolome and clinicians with a readily available resource to learn more about healthy human urine.

## Figures and Tables

**Figure 1 diagnostics-12-02184-f001:**
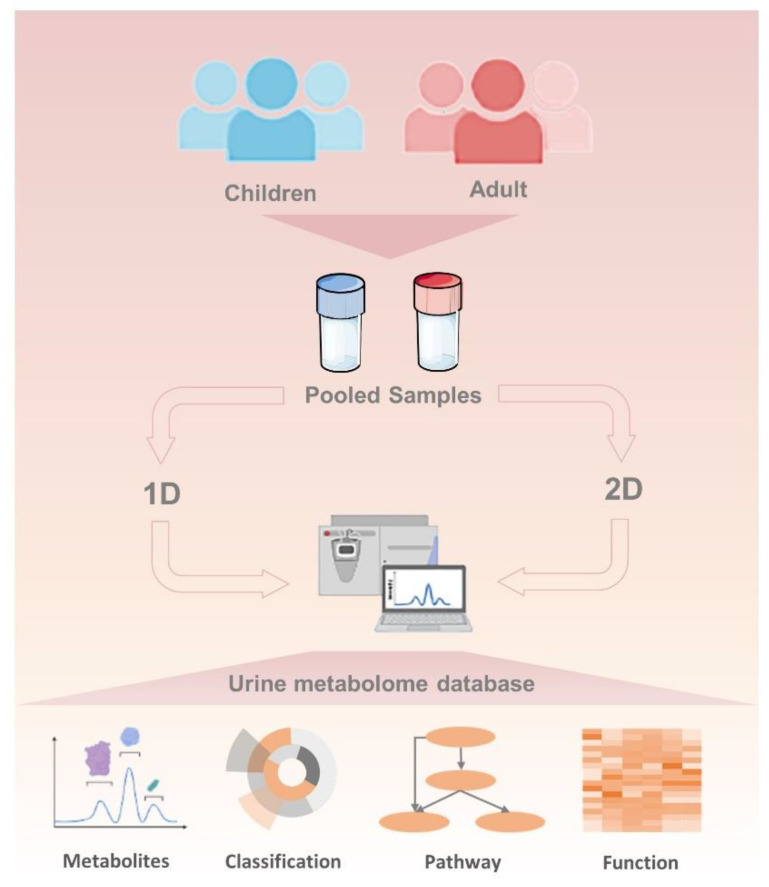
**The workflow of this study:** 1D means one-dimensional separation, and 2D means two-dimensional separation.

**Figure 2 diagnostics-12-02184-f002:**
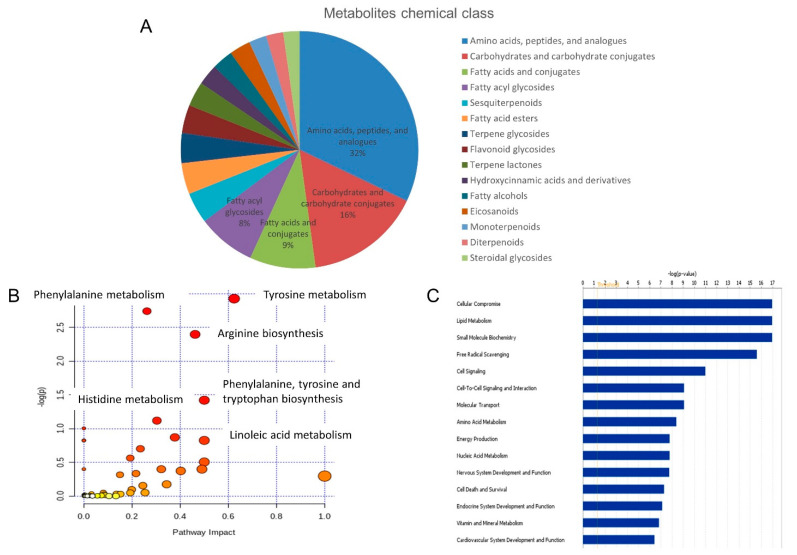
**Urine metabolome profile analysis.** (**A**) Chemical class of the identified metabolites. (**B**) Pathway analysis of the identified metabolites. (**C**) Functional annotation of the identified metabolites.

**Figure 3 diagnostics-12-02184-f003:**
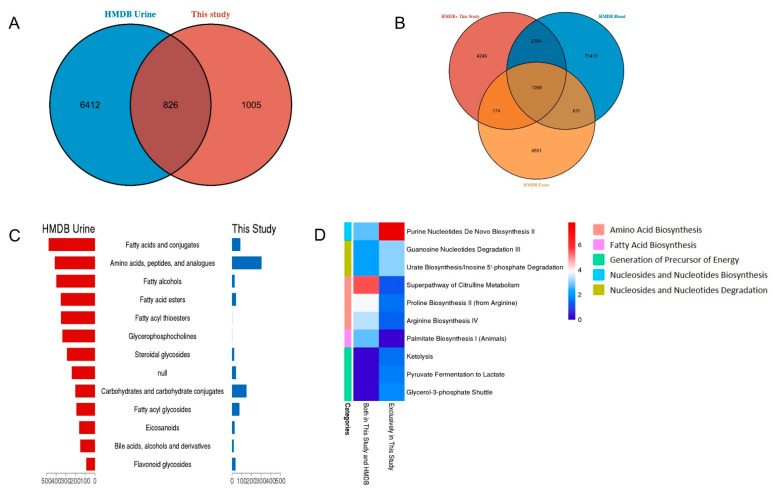
**Comparison between this study and HMDB.** (**A**) The overlap of this study and HMDB. (**B**) Comparison of the combined urine database with HMDB blood and fecal database. (**C**) The top sub class of annotated metabolites in this study and HMDB. (**D**) Pathway comparative analysis of this study and HMDB.

**Figure 4 diagnostics-12-02184-f004:**
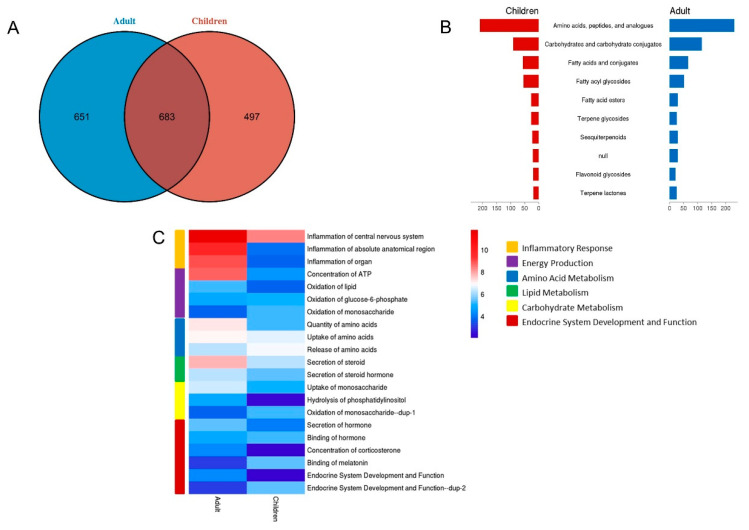
**Urine metabolomics comparison between children and adults.** (**A**) The overlap between the children’s and adult’s urinary metabolites. (**B**) Chemical class of the urinary metabolites from children and adults. (**C**) Functional comparison of the urinary metabolites from children and adults.

**Figure 5 diagnostics-12-02184-f005:**
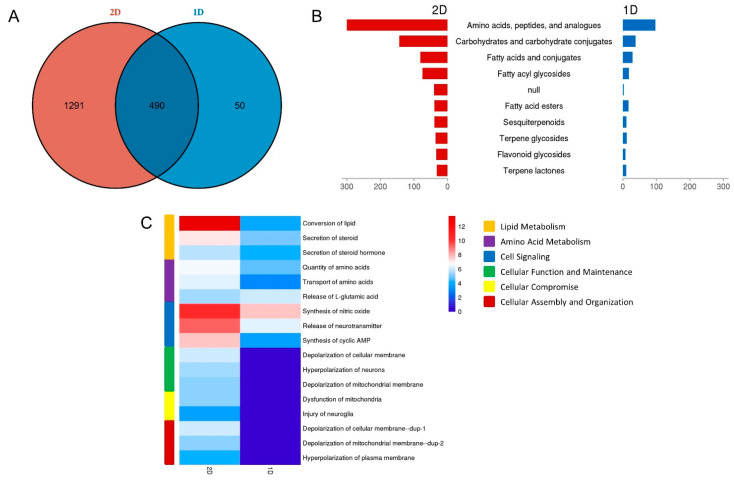
**Urine metabolomics comparison between 1D and 2D separation strategy.** (**A**) The overlap between the 1D and 2D. (**B**) Chemical class of the urinary metabolites from 1D and 2D. (**C**) Functional comparison of the urinary metabolites from 1D and 2D.

**Figure 6 diagnostics-12-02184-f006:**
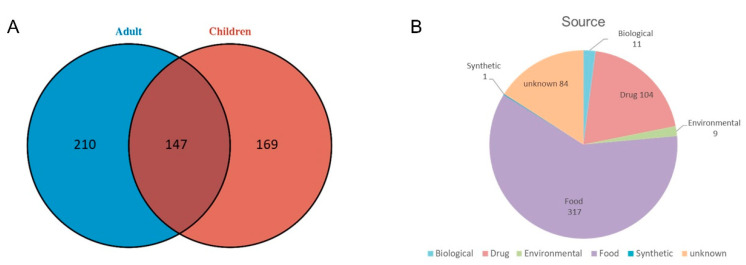
**Detection of exogenous metabolites.** (**A**) The overlap among the exogenous metabolites identified in children and adults. (**B**) The source of exogenous metabolites.

**Table 1 diagnostics-12-02184-t001:** Metabolites identified in urine from endogenous.

Method	Children	Adults	Total
1D	386	307	540
2D	1032	1240	1781
Total	1180	1334	1831

**Table 2 diagnostics-12-02184-t002:** Metabolites identified in urine from exogenous.

Method	Children	Adults	Total	Total
1D	82	70	124	124
2D	274	330	476	476
Total	316	357	526	526

## Data Availability

The mass spectrometry metabolomics data have been deposited in the integrated proteome resources (iProX) with the dataset identifier IPX0004220000. We confirm that the authors are accountable for all aspects of the work (if applied, including full data access, the integrity of the data, and the accuracy of the data analysis) in ensuring that questions related to the accuracy or integrity of any part of the work are appropriately investigated and resolved.

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
