# Peer review of "A Comprehensive 2D-LC/MS/MS Profile of the Normal Human Urinary Metabolome"

_diagnostics, 2022, doi:10.3390/diagnostics12092184_

Round 1
Reviewer 1 Report
I read with a really great interests paper entitled “Comprehensive 2D-LC/MS/MS profile of normal human urinary metabolome”, in which the authors described profiling of body fluids and its role in monitoring and discovering metabolic markers of diseases.
I have some comments to this paper.
It would be nice to have precisely described all inclusion and exclusion criteria. Did you include participants with some dietary restrictions? How about patients with enzymes deficiencies, i.e. phenylalanine hydroxylase deficiency?
Did you include pregnant or breastfeeding women?
What was the smoking status of participants in your study?
How about renal insufficiency in your population?
The authors should also add all potential limitations of study.
Reviewer 2 Report
In this manuscript, the authors used LC-MS/MS (1D/2D) to profile urine samples obtained from healthy individuals. Several comparison including 1D vs. 2D methods, adults vs. children, etc. were performed. Various metabolite functions were also investigated. The authors should address the following points before consideration:
1. The authors should include representative LC-MS/MS profiles (real data).
2. Why was fractionation by HPLC necessary before analyses?
3. Could the authors clarify the identity and source of QC sample? It is not clear to me.
4. A huge number of new metabolites (1005) were found in this study. Any plausible explanation? Is this population-specific?
5. In LC-MS/MS analysis section, the authors should clearly describe how 1D vs. 2D acquisition methods were carried out.
Reviewer 3 Report
The manuscript brings high quality content highlighting aspects that will provide important data for both clinicians (practitioners) and lab workers.
Author Response
Thanks for the comments.
Round 2
Reviewer 2 Report
The authors' response is satisfactory.